# The Role of Vitamin D Supplementation on Airway Remodeling in Asthma: A Systematic Review

**DOI:** 10.3390/nu15112477

**Published:** 2023-05-26

**Authors:** Laila Salameh, Walid Mahmood, Rifat Hamoudi, Khulood Almazrouei, Mahesh Lochanan, Suheyl Seyhoglu, Bassam Mahboub

**Affiliations:** 1Dubai Academic Health Corporation, Rashid Hospital, Dubai P.O. Box 4545, United Arab Emirates; xwalidx@gmail.com; 2Research Institute of Medical and Health Sciences, University of Sharjah, Sharjah P.O. Box 27272, United Arab Emirates; rhamoudi@sharjah.ac.ae (R.H.); maravindalochanan@dha.gov.ae (M.L.); sseyhoglu@dha.gov.ae (S.S.); drbassam_mahboub@yahoo.com (B.M.); 3Division of Surgery and Interventional Science, University College London, London HA4 4LP, UK; 4Zayed Military Hospital, Abu Dhabi P.O. Box 72763, United Arab Emirates; k.almazrouei8308@gmail.com

**Keywords:** asthma, vitamin D, airway remodeling, inflammation, smooth muscle cell

## Abstract

Asthma is a common chronic respiratory disease that affects millions of people worldwide, and its prevalence continues to increase. Vitamin D has been proposed as a potential environmental factor in asthma pathogenesis, due to its immunomodulatory effects. This systematic review aimed to evaluate the effect of vitamin D supplementation in order to prevent airway remodeling in asthmatic patients. Four electronic databases, namely PubMed, Embase, Clinical trails.gov, and CINAHL, were thoroughly searched to conduct a comprehensive literature review. The International Prospective Register of Systematic Reviews (CRD42023413798) contains a record of the registered protocol. We identified 9447 studies during the initial search; 9 studies (0.1%) met the inclusion criteria and were included in the systematic review. All included studies were experimental studies that investigated the impact of vitamin D supplementation on airway remodeling in asthma. The studies included in this review suggest that vitamin D inhibits airway smooth muscle cell contraction and remodeling, reduces inflammation, regulates collagen synthesis in the airways, and modulates the action of bronchial fibroblasts. However, one study suggests that TGF-β1 can impair vitamin D-induced and constitutive airway epithelial host defense mechanisms. Overall, vitamin D appears to have a potential role in the prevention and management of asthma.

## 1. Introduction

Asthma is a chronic respiratory disease that affects around 334 million people globally, and it is attributed to 250,000 deaths yearly. It is estimated that the prevalence of asthma will increase by another 100 million by 2025 as asthma incidence continues to grow [1,2]. Asthma is characterized by airway inflammation, airway hyperresponsiveness, and airflow limitation, which can lead to respiratory symptoms such as coughing, wheezing, and shortness of breath [3]. The pathogenesis of asthma is complex and involves various genetic, environmental, and immunological factors. One of the potential environmental factors that has been studied in relation to asthma is vitamin D [4].

Vitamin D is a type of vitamin that can dissolve in fat and is naturally produced by the body’s skin when it is exposed to sunlight. Additionally, it can be obtained from specific food sources such as egg yolks, dairy products, and fatty fish [5,6]. Vitamin D has been shown to possess immunomodulatory properties and is involved in the regulation of various immune cells, including T cells and dendritic cells.

Over the past twenty years, numerous research groups have explored the connection between vitamin D and asthma pathogenesis. Their findings indicate that asthma patients with insufficient levels of vitamin D tend to have more severe symptoms and poorer lung function [7,8,9]. Various epidemiological and meta-analysis studies have indicated that children with low levels of vitamin D in their blood are more likely to develop asthma and experience more exacerbations, worsening asthma symptoms, and reduction in lung function if they already have asthma [10,11,12,13]. Additionally, there appears to be a correlation between low maternal vitamin D intake and levels during pregnancy and an increased likelihood of wheezing in children [10,14]. The evidence suggests that vitamin D affects both innate and adaptive immune system cells and structural cells in the airways, and that deficiency can promote inflammation while supplementation can alleviate these effects [8,15,16].

Airway remodeling in asthma is characterized by an increase in the thickness of the airway smooth muscle (ASM) layer, which is caused by both hypertrophy and hyperplasia of ASM cells [17]. Many pro-inflammatory and pro-fibrotic mediators are known to be produced by bronchial smooth muscle cells (BSMCs), highlighting their significance as a source of these molecules [18]. Hence, the excessive proliferation of smooth muscle cells and their heightened secretion of inflammatory mediators may lead to amplified airflow obstruction and extracellular matrix (ECM) deposition, ultimately resulting in fibrosis in individuals affected by asthma [19].

The impact of vitamin D on inflammatory mechanisms has been investigated in non-allergic asthma as well. Various studies have extensively explored the underlying mechanisms of airway tissue remodeling in asthma, with a focus on the structural cells responsible for this irreversible process [9,17,20]. Current scientific knowledge confirms that structural cells, such as epithelial and ASM cells, play a role in the development of asthma through complex interactions with inflammatory lymphocytes. The degradation of airway epithelial integrity and the process of epithelial–mesenchymal transition (EMT) during airway remodeling are substantial factors in the pathogenesis of asthma [21].

The aim of the present systematic review is to synthesize and critically appraise the literature on studies that investigated the effect of vitamin D on airway remodeling in patients with asthma.

## 2. Materials and Methods

### 2.1. Study Protocol and Registration

The systematic review was conducted following the guidelines set forth by the Cochrane Collaboration and the Preferred Reporting Items for Systematic Reviews (PRISMA) statement. We registered our study in the PROSPERO database https://www.crd.york.ac.uk/PROSPERO/ (accessed on 25 April 2023) and were assigned the registration ID (CRD42023413798).

### 2.2. Search Strategy

The review search strategy was begun by conducting a systematic search of the PubMed, Embase, Clinical trails.gov, and CINAHL databases using MeSH (Medical Subject Headings) using the terms “vitamin D” OR “25-hydroxyvitamin D” OR “vit d” and asthma and airway remodeling (Appendix A). Additionally, manual searches were performed, followed by cross-referencing relevant articles to ensure a comprehensive review of the literature. We analyzed the chosen studies using Review Manager 5.3 software, which was developed by the Cochrane Library, and we established a significance level of 5% (*p* < 0.05).

### 2.3. Eligibility Criteria

The study’s inclusion criteria, which investigated the relationship between vitamin D supplementation and asthma airway remodeling, were limited to randomized controlled human trials or experimental studies published in the English language over the past decade (1 January 1990 to 28 February 2023) and adults > 18 years old. Exclusion criteria were studies involving children (<18 years old) or maternal populations. Articles such as review and editorial pieces, letters to editors, brief communications, short communications, personal opinions, and commentaries were excluded from consideration for this study.

### 2.4. Quality Assurance and Data Extraction

Two independent reviewers (L.S. and B.M.) objectively reviewed the materials, and a consensus was reached by ensuring that they matched the inclusion criteria and MeSH terms. A risk-of-bias assessment was carried out using the Cochrane risk-of-bias tool for randomized trials (RoB 2) (Appendix A), in accordance with the selected study criteria [22]. The risk of bias assessment figure used robvis for traffic plotting (https://mcguinlu.shinyapps.io/robvis/) (accessed on 25 April 2023) [23]. To ensure consistency in the selection of studies for inclusion, Cohen’s kappa was utilized to assess inter-rater reliability between the two reviewers (approximately 20 titles and abstract screenings and 10 full-text screenings were conducted). Cohen’s kappa coefficient was used to determine the level of agreement between the two reviewers, with values less than 0.20 indicating slight agreement, 0.21–0.40 indicating fair agreement, 0.41–0.60 indicating moderate agreement, 0.61–0.80 indicating substantial agreement, and 0.81–1.00 indicating perfect agreement. Any discrepancies between the reviewers were resolved through an iterative consensus process.

### 2.5. Data Variables

The extracted data were organized based on the following criteria: first author’s name and year of publication, study design, country of origin, cell types used, vitamin D intervention details, outcome measurements of airway remodeling, and methods used to assess airway remodeling. We attempted to identify randomized controlled trials and experimental studies investigating the effects of interventions on airway remodeling. All studies that measured airway remodeling were experimental studies rather than randomized controlled trials.

## 3. Results

### 3.1. Study Selection

A comprehensive literature search was performed using PubMed, Embase, Clinical trails.gov, and CINAHL, as well as a manual search, which retrieved a total of 9447 studies related to vitamin D, 25-hydroxyvitamin D, vit d, asthma, and airway remodeling. After removing duplicate studies, 6650 remained; among those studies, the titles and abstracts of 2472 were screened, resulting in the exclusion of 2391 studies. The full text of the remaining 81 articles was reviewed, and 72 were excluded based on pre-specified criteria (Figure 1). The excluded studies used animal models, children, or prenatal populations (54 articles), were narrative reviews (6 articles), did not measure the effect of vitamin D on airway remodeling as an outcome of interest (10 articles), or were duplicates (2 articles). Of the remaining articles, nine articles were included for narrative systematic review.

### 3.2. Characteristics of the Studies

Out of 9447 studies identified during the initial search, 9 studies (0.1%) met the inclusion criteria and were included in the systematic review. All the included studies were in vitro experimental studies conducted between 2007 and 2021.

Four of the included studies were from China [24,25,26,27], three were from the USA [28,29,30], and two were from Canada [31,32]. Five studies used human airway smooth muscle cells [26,27,28,31,32], while the other three studies used bronchial fibroblast [24,29,32]. Two studies measured airway remodeling directly [27,28], while the other six studies did not measure remodeling directly. Instead, they investigated the underlying mechanisms impairing airway epithelial or fibroblast cells, which will lead to airway remodeling [24,26,29,30,31,32], as seen in Table 1.

### 3.3. Risk of Bias Assessment

The risk of bias assessment for each included clinical study was conducted using the Cochrane RoB 2 tool and presented in a traffic light plot (Appendix A) [23]. The traffic light plot reports five risk of bias domains: D1, bias arising from the randomization process; D2, bias due to deviations from intended intervention; D3, bias due to missing outcome data; D4, bias in the measurement of the outcome; and D5, bias in the selection of the reported result; the yellow circle indicates some concerns on the risk of bias and the green circle represents a low risk of bias. The risk of bias assessment showed that all of the studies included in this review were at low risk of bias, indicating that the quality of the evidence is high.

### 3.4. Vitamin D Inhibits Airway Smooth Muscle Cell Contractions and Remodeling

Of the nine reviewed studies, five studies suggested that vitamin D may have a protective effect against airway smooth muscle cell contraction and remodeling in asthma [25,26,28,30,31]. Bossé et al., 2007 [31] found that treatment with 1,25-dihydroxyvitamin D3 (the active form of vitamin D) induced autocrine (self-stimulating), contractility, and remodeling processes in bronchial smooth muscle cells [31]. Specifically, the authors found that vitamin D increased the expression of genes involved in muscle contraction, such as smooth muscle alpha-actin, and in remodeling, such as matrix metalloproteinase-9. While Song et al., 2007 [27] examined the effects of 1,25-dihydroxyvitamin D3 on passively sensitized human airway smooth muscle cells (cells that have been exposed to allergens) [27]. They found that vitamin D reduced the contractile response of these cells to acetylcholine, a neurotransmitter that promotes muscle contraction. The authors suggested that this effect may be mediated by decreased intracellular calcium levels, which are necessary for muscle contraction. On the other hand, Damera et al., 2009 [30] investigated the mechanisms by which vitamin D inhibits the growth of human airway smooth muscle cells. They found that vitamin D reduced the expression of genes involved in cell cycle progression, such as cyclin D1, and induced the phosphorylation of retinoblastoma protein and checkpoint kinase 1, which are key regulators of the cell cycle. These effects ultimately led to decreased cell proliferation [30]. Moreover, Britt et al., 2016 [28] examined the effects of vitamin D on inflammation-induced contractility and remodeling of asthmatic human airway smooth muscle. They found that vitamin D reduced the expression of genes involved in inflammation, such as IL-6 and IL-8, and inhibited the contraction of airway smooth muscle cells in response to inflammatory stimuli [28].

Overall, these studies suggest that vitamin D may have a protective effect against airway smooth muscle cell contraction and remodeling in asthma. Finally, Kim, Sung-Ho et al. (2017) [25] reported that active vitamin D3 (1,25(OH)2D3) inhibited VEGF-induced ADAM33 expression and proliferation in human airway smooth muscle cells. The researchers suggested that this finding has implications for asthma treatment, as ADAM33 is involved in airway remodeling, a key feature of asthma. They proposed that vitamin D3 supplementation may have the potential as an adjunct therapy for asthma by reducing airway remodeling [25].

### 3.5. Vitamin D Can Reduce Inflammation and Regulate Collagen Synthesis in the Airways

Two studies suggested that vitamin D can reduce inflammation and regulate collagen synthesis in the airways, as seen in the studies by Song et al., 2013 [26] and Jin et al., 2021 [24]. Inflammation is a key feature of asthma and is characterized by the infiltration of immune cells and the production of inflammatory mediators such as cytokines. Song et al., 2013 investigated the effect of vitamin D on nuclear factor kappa B (NF-κB), a transcription factor that plays a crucial role in the regulation of inflammatory responses. The authors found that vitamin D reduced the activation of NF-κB in passively sensitized human airway smooth muscle cells, thereby suppressing the expression of pro-inflammatory cytokines such as IL-6 and IL-8. They suggested that this effect may be mediated by the stabilization of inhibitor IκBα, a protein that prevents NF-κB from entering the nucleus and activating gene expression [26].

Collagen is a major component of the extracellular matrix in the airways and is responsible for maintaining the structural integrity of the airway wall. However, in asthma, collagen synthesis can be dysregulated, leading to excessive collagen deposition and airway remodeling. Jin et al., 2021 [24] investigated the effect of vitamin D on collagen synthesis in human lung fibroblasts, which are cells that are responsible for producing collagen in the airways. The authors found that vitamin D reduced the expression of collagen type I and the activity of enzymes involved in collagen synthesis, such as protein arginine methyltransferase 1 (PRMT1). They suggested that this effect may be mediated by the inhibition of signal transducer and activator of transcription 3 (STAT3), a transcription factor that regulates collagen synthesis [24]. Taken together, these studies suggest that vitamin D may have anti-inflammatory and anti-fibrotic effects in the airways, which may help to reduce the severity of asthma and prevent airway remodeling.

### 3.6. Vitamin D Can Modulate the Action of Bronchial Fibroblasts

The study by Plesa et al., 2020 [32] investigated the effect of vitamin D on human asthmatic bronchial fibroblasts, which are cells that are involved in airway remodeling and contribute to asthma worsening [32].

The authors found that treatment with vitamin D (specifically 1,25(OH)2D3) reduced the proliferation and migration of bronchial fibroblasts in a dose-dependent manner, suggesting that vitamin D can inhibit the growth of these cells. They also found that vitamin D reduced the expression of genes involved in extracellular matrix remodeling, such as collagen type I and matrix metalloproteinase 2 (MMP2), indicating that vitamin D can modulate the action of bronchial fibroblasts. The authors further investigated the mechanisms by which vitamin D exerts its effects on bronchial fibroblasts and found that it inhibits the activity of several signaling pathways that are involved in cell proliferation and migration, such as ERK1/2 and Akt. They also found that vitamin D upregulates the expression of genes involved in cell cycle arrest, such as p21 and p27, suggesting that vitamin D can induce cell cycle arrest in bronchial fibroblasts [32].

Overall, this study suggests that vitamin D can modulate the action of bronchial fibroblasts by inhibiting their proliferation and migration, and by reducing the expression of genes involved in extracellular matrix remodeling. These findings may have important implications for the prevention and treatment of asthma, as airway remodeling is a key feature of the disease.

### 3.7. TGF-β1 Can Impair Vitamin D-Induced and Constitutive Airway Epithelial Host Defense Mechanisms

The study by Schrumpf et al., 2020 [29] investigated the relationship between two important molecules in the airways: vitamin D and TGF-β1. Vitamin D is known to play a role in regulating immune responses in the airways and promoting host defense against respiratory pathogens, while TGF-β1 is a cytokine that is involved in the development of asthma and other respiratory diseases. The authors found that treatment with TGF-β1 impaired the ability of airway epithelial cells to induce the expression of genes involved in host defense when stimulated with vitamin D [29]. This suggests that TGF-β1 can inhibit the action of vitamin D in promoting host defense mechanisms in the airways. The authors further investigated the mechanisms underlying this effect and found that TGF-β1 interferes with the signaling pathway activated by vitamin D, specifically by reducing the expression of the vitamin D receptor (VDR) and suppressing the activity of the transcription factor NF-κB. These effects lead to a reduced expression of genes involved in host defense and an impaired ability of the airway epithelial cells to fight off respiratory pathogens.

Overall, this study suggests that TGF-β1 can impair the action of vitamin D in promoting host defense mechanisms in the airways by interfering with the signaling pathway activated by vitamin D, and reducing the expression of key genes involved in host defense. These findings may have important implications for the development of new therapies for respiratory diseases such as chronic obstructive pulmonary disease (COPD), as targeting the interaction between vitamin D and TGF-β1 may be a promising approach for improving airway host defense.

## 4. Discussion

In our study, the outcomes of the included studies on the identification effect of vitamin D on asthma airway remodeling suggested that vitamin D has a potential role in regulating several key processes involved in respiratory diseases, such as airway smooth muscle cell contraction and remodeling, inflammation, collagen synthesis, and the action of bronchial fibroblasts. Specifically, the studies incorporated in this review offer proof that vitamin D can inhibit contraction and remodeling of the smooth muscle cell in the airway, which are key features of asthma and other respiratory diseases. The studies by Bossé et al. (2007) [31], Song et al. (2007) [27], Damera et al. (2009) [30], Britt et al. (2016) [28], and Kim, Sung-Ho et al. (2017) [25] all demonstrate the ability of vitamin D to modulate these processes, suggesting that it may be a promising therapeutic target for respiratory diseases. These findings are in line with other studies that have demonstrated that lower levels of serum vitamin D are associated with increased airway smooth muscle (ASM) mass, airway hyperresponsiveness (AHR), poorer asthma control, and increased exacerbations in children, adolescents, and adults [12,13,33]. These studies suggest that vitamin D can be a potential therapeutic agent for asthma and other airway diseases. By inhibiting airway smooth muscle cell contraction and remodeling, vitamin D may reduce airway hyperresponsiveness and improve airway function. However, further studies are necessary to investigate the underlying mechanisms through which vitamin D exerts its effects on airway smooth muscle cells. Additionally, clinical trials are needed to determine the efficacy of vitamin D supplementation in treating airway diseases.

In addition, other studies, such as those of Song et al. (2013) [26] and Jin et al. (2021) [24], support the notion that vitamin D can reduce inflammation and regulate collagen synthesis in the airways, which are critical processes in airway remodeling. Previously, it was believed that airway remodeling was a consequence of chronic inflammation over a long period of time. However, recent evidence indicates that the process of remodeling begins in early childhood, even before the age of three years [34]. Observational studies suggest that vitamin D may have a protective role in severe asthma. The ability to modulate the immune system with vitamin D has been associated with asthma management, and there is growing evidence to support the role of the vitamin D pathway in regulating immune function. The vitamin D receptor (VDR) has been found in almost all immune cells, including macrophages, dendritic cells, T-cells, and B-cells [6,35,36,37,38]. In mouse models, studies have shown that VDR knockout mice do not develop experimental asthma, suggesting that vitamin D is required for the generation of T-helper (Th)2-driven inflammation in the airways [39].

Furthermore, Plesa et al. (2020) [32] observed that vitamin D can modulate the action of bronchial fibroblasts, which play a crucial role in the development of airway remodeling. They found that treatment with 1,25(OH)2D3 led to a reduction in the proliferation of asthmatic bronchial fibroblasts, as well as a decrease in the production of extracellular matrix proteins and pro-inflammatory cytokines. Several studies have reported on the anti-fibrotic effects of vitamin D in various disease models. In VDR knockout mice, an increase in inflammatory cell infiltration, upregulation of metalloproteinases, and phosphoacetylation of NF-κB were observed in the lung, which was associated with emphysema and a decline in lung function along with the formation of lymphoid aggregates [40]. This highlights the importance of vitamin D in regulating these processes. Moreover, in tuberculosis, 1,25(OH)2D3 was shown to suppress the production of MMPs while enhancing TIMP-1 levels [41]. In the cell line of human squamous carcinoma, vitamin D3 was found to reduce the production of both MMP-9 and MMP-13 mRNA and proteins in a manner that is dependent on the dosage applied [42]. Similarly, in a model of Crohn’s disease, a vitamin D analog was able to weaken the pro-fibrotic reaction of colonic myofibroblasts when exposed to elevated levels of matrix stiffness [43]. These findings suggest that vitamin D may have a potential therapeutic effect in preventing fibrosis in asthma.

However, it’s important to note that the review also highlights the potential for TGF-β1 to impair vitamin D-induced and constitutive airway epithelial host defense mechanisms, as shown by Schrumpf et al. (2020) [29]. The deposition of collagen is excessively increased when fibroblasts are activated by tumor growth factor-beta-1 (TGF-β1), which is a pro-fibrotic cytokine. This activation plays a crucial role in asthma, as TGF-β is synthesized by various cells such as macrophages, lymphocytes, eosinophils, fibroblasts, and airway epithelial cells [44]. Additionally, Th2-derived cytokines, including IL-4, are essential in airway remodeling. TGF-β also induces the expression of TIMP-1, and this process is dependent on Th2 [45,46]. This suggests that the relationship between vitamin D and airway remodeling is complex, and further research is needed to entirely identify the mechanisms involved.

## 5. Limitations

One limitation of this review is that it only examined the effects of vitamin D on adults and did not consider the potential impact on children and pregnant women, as the patterns and severity of airway remodeling may vary across different populations with asthma.

## 6. Conclusions

In conclusion, the studies reviewed provide evidence that vitamin D has beneficial effects on the airways, including inhibition of airway smooth muscle cell contraction and remodeling, reduction in inflammation, and regulation of collagen synthesis. Some studies suggest that vitamin D can also modulate the action of bronchial fibroblasts. However, it is important to note that one study suggests that TGF-β1 can impair vitamin D-induced and constitutive airway epithelial host defense mechanisms. Overall, the findings suggest that vitamin D may have a potential role in the prevention and management of asthma.

## Figures and Tables

**Figure 1 nutrients-15-02477-f001:**
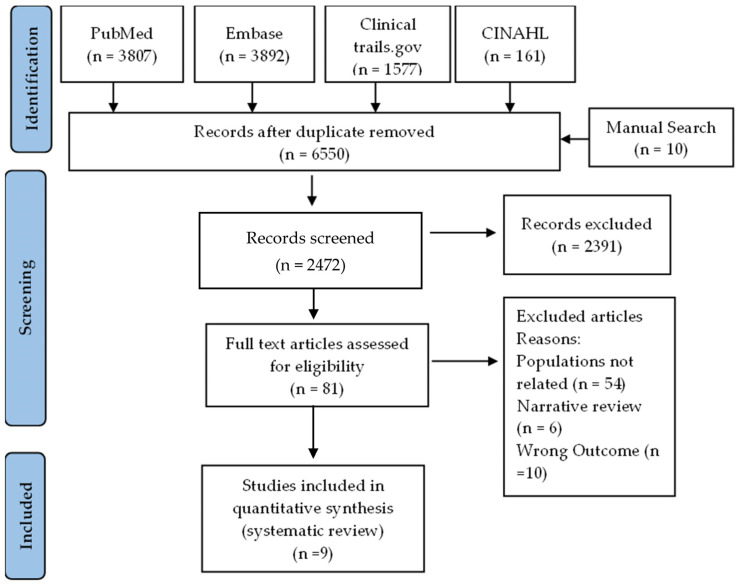
PRISMA flowchart depicting the study selection process according to inclusion and exclusion criteria.

**Table 1 nutrients-15-02477-t001:** Included studies’ general characteristics.

Author/Year	Study Design	Country	Cell Type	Intervention	Outcomes
Bossé Y et al. (2007) [31]	In vitro experimental study	Canada	BSMC	Stimulation with 1α,25-dihydroxy-vitamin D3 at dose of 100 nM	Autocrine, contractility, and remodeling processes.
Song Y et al. (2007) [27]	In vitro experimental study	China	HASMCs	Treatment with 1,25-(OH)2D3 (a vitamin D analog) at dose of 10 nM	Changes in cell proliferation, intracellular calcium levels, and cytokine production in response to treatment with 1,25-(OH)2D3.
Damera G. (2009) [30]	In vitro experimental study	USA	HASMCs	Treatment with vitamin D (1,25-dihydroxyvitamin D3) at dose of 1 µM	Vitamin D has been found to hinder the growth of human airway smooth muscle cells by promoting the phosphorylation of retinoblastoma protein and checkpoint kinase 1. This effect occurs in response to growth factors such as platelet-derived growth factor (PDGF), epidermal growth factor (EGF), and insulin-like growth factor-1 (IGF-1).
Song Y et al. (2013) [26]	In vitro experimental study	China	HASMCs	Treatment with 1,25-dihydroxyvitamin D3 at dose of 10^−7^ m	Expression and phosphorylation of inhibitor IκBα, expression of NF-κB p65 subunit, mRNA stability, and NF-κB DNA binding activity.
Britt RD Jr et al. (2016) [28]	In vitro experimental study	USA	HASMCs	Treatment with vitamin D at dose of 100 nM	In asthmatic patients, Vitamin D was observed to decrease the inflammation-induced contractility and remodeling of airway smooth muscle (ASM) cells.
Kim, Sung-Ho et al. (2017) [25]	In vitro experimental study	China	HASMCs	Treatment with vitamin D. Various doses of 1,25(OH)2D3 used varied from 0 to 100 nM	The study used human airway smooth muscle cells (HASMCs) in vitro and evaluated the effect of active vitamin D3 on VEGF-induced ADAM33 expression and proliferation of these cells, which are associated with airway remodeling in asthma.
Plesa M et al. (2020) [32]	In vitro experimental study	Canada	human asthmatic bronchial fibroblasts	Treatment with 1,25-dihydroxyvitamin D3 with two different doses, 50 nM or 100 nM	1,25(OH)(2)D(3) was able to reduce the expression of extracellular matrix proteins and inhibit fibroblast proliferation and migration in asthmatic bronchial fibroblasts, potentially highlighting a role for vitamin D in the prevention of airway remodeling in asthma.
Schrumpf JA. (2020) [29]	In vitro experimental study	USA	human asthmatic bronchial fibroblasts	Treatment with 1,25-dihydroxyvitamin D3 at dose of 100 nM	TGF-β1 impairs the ability of vitamin D to induce and maintain airway epithelial host defense mechanisms. Specifically, TGF-β1 inhibited vitamin D-induced expression of the antimicrobial peptide cathelicidin and reduced vitamin D receptor (VDR) expression.
Jin A et al. (2021) [24]	In vitro experimental study	China	HBFs	Treatment with calcitriol and dose used at 10 nM	TSLP-induced collagen type-I synthesis through STAT3 and PRMT1.

Abbreviations: Bronchial smooth muscle cells (BSMC); human airway smooth muscle cells (HASMCs); human asthmatic bronchial fibroblasts (HABFs); human bronchial fibroblasts (HBFs); thymic stromal lymphopoietin (TSLP); vitamin D receptor (VDR); insulin-like growth factor 1 (IGF1); and epidermal growth factor (EGF).

## Data Availability

The data presented in this study are available upon request from the corresponding author.

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
