# Peer review of "The Role of Vitamin D Supplementation on Airway Remodeling in Asthma: A Systematic Review"

_nutrients, 2023, doi:10.3390/nu15112477_

Round 1

Reviewer 1 Report

Overall Summary. This manuscript “The role of Vitamin D supplementation on Airway Remodeling 2 in Asthma: A Comprehensive Systematic Review” is a review aiming to evaluate the effect of vitamin D supplementation on airway remodeling in asthma. Authors only included  in vitro experimental studies , and has excluded studies in children or maternal population leading to a generalizability issue. Please see my specific comments below:

Title: Please delete “comprehensive”

Abstract

The total number of final included papers needs to be mentioned in the abstract. And line 17:  how many “studies” had this conclusion?

After reading the paper, in the abstract, authors need to be clear that all included studies were in vitro, no human studies were included.

Introduction

Line 45: if there is only one reference, I wouldn’t call it “various”. Please add more references as indicated.

Methods

Line 78: Again, it’s unclear to me why authors exclude studies in children or maternal populations. Given this kind of study would be rare, it will be important to include in the review to inform the audience if there is any.

Line 90. Authors stated “All studies that measured airway remodeling were experimental studies rather than randomized controlled trials.” However, in the introduction, the authors mentioned “Several clinical trials have investigated the effect of vitamin D supplementation on asthma outcomes, including airway remodeling.” These are opposite points, please revise.

Results

Line 100: the description of the criteria for exclusion, such as “populations not related” or “wrong outcome”, is unclear. Please revise them to make them more concise.

Line 109: there is a space for “ in vitro”

Line 111: I believe “Pennsylvania” is a State of the USA

Discussion

what do the authors recommend for future studies? What are the current gaps? And what are the clinical implications for the findings?

Please also add a limitation section

Conclusion

Line 280: Since all studies were in vitro, the conclusion needs to step down.

English is ok, but some edits are needed to make sure it's concise.

Author Response

Review 1 :

Overall Summary. This manuscript “The role of Vitamin D supplementation on Airway Remodeling 2 in Asthma: A Comprehensive Systematic Review” is a review aiming to evaluate the effect of vitamin D supplementation on airway remodeling in asthma. Authors only included  in vitro experimental studies  and has excluded studies in children or maternal population leading to a generalizability issue. Please see my specific comments below:

Title: Please delete “comprehensive”

Response 1: We thank the reviewer for pointing this out, we have corrected the title as suggested , kindly check line 3 .

Abstract

The total number of final included papers needs to be mentioned in the abstract. And line 17:  how many “studies” had this conclusion?

After reading the paper, in the abstract, authors need to be clear that all included studies were in vitro, no human studies were included.

Response 2: We thank the reviewer for pointing this out. Addendum done as suggested, kindly check lines 15-16.

Introduction

Line 45: if there is only one reference, I wouldn’t call it “various”. Please add more references as indicated.

Response 3: We thank the reviewer for pointing this out. More studies added.

Methods

Line 78: Again, it’s unclear to me why authors exclude studies in children or maternal populations. Given this kind of study would be rare, it will be important to include in the review to inform the audience if there is any.

Response 4: We thank the reviewer for pointing this out. While airway remodeling can occur in all populations with asthma, the patterns and severity of remodeling may differ between adults, children, and pregnant women with asthma. So, we decided it to focus on adult.

Line 90. Authors stated, “All studies that measured airway remodeling were experimental studies rather than randomized controlled trials.” However, in the introduction, the authors mentioned “Several clinical trials have investigated the effect of vitamin D supplementation on asthma outcomes, including airway remodeling.” These are opposite points, please revise.

Response 5: We thank the reviewer to bring this point to notice. We revised the statement and we corrected to (Several experimental studies ) in the introduction. Kindly check line 53.

Results

Line 100: the description of the criteria for exclusion, such as “populations not related” or “wrong outcome”, is unclear. Please revise them to make them more concise.

Response 6: We thank the reviewer to bring this point to notice. We corrected “populations not related” to “The excluded studies were used animal models, children, or prenatal populations”   or “wrong outcome” to “ did not measure airway remodeling as an outcome of interest”. Kindly check lines 112-114.

Line 109: there is a space for “ in vitro”

Response 7: We thank the reviewer for spotting this error. We have corrected add space between “in vitro”.

Line 111: I believe “Pennsylvania” is a State of the USA

Response 8: We thank the reviewer to bring this point to notice. We corrected and added the study among USA studies kindly line 123.

Discussion

what do the authors recommend for future studies? What are the current gaps? And what are the clinical implications for the findings?

Please also add a limitation section

Response 9: We thank the reviewer for the comments .We recommended in the line 249-252 & 292-294 in the discussion further studies. And as clinical implications we mentioned it in line 281-283. Limitation added ,Kindly check line 295-299.

Conclusion

Line 280: Since all studies were in vitro, the conclusion needs to step down.

Response 10 :We thank the reviewer for the comment.

Reviewer 2 Report

The manuscript by Salameh et al. examines the effect of vitamin D supplementation on airway remodeling in asthma. The authors conducted a comprehensive search of electronic databases and identified experimental studies and randomized controlled trials.

The abstract could benefit from more specific information about the number of studies included in the review and the sample sizes of those studies.

Authors should rephrase the abstract. They should be clearer in the material and methods section of the abstract, indicating the databases used, study selection criteria, study population, number of articles finally included in the review.

It would be helpful to include any limitations or potential biases of the included studies or the review process.

The conclusion could be more specific in terms of recommendations for clinical practice or future research directions.

Introduction.

Overall, the introduction is well written but could be improved by providing more context on the importance of airway remodelling in asthma and how it relates to the overall pathogenesis of the disease. In addition, the introduction could benefit from a clearer statement of the research question or objective of the systematic review, which could help readers better understand the purpose of the study.

Material and methods

The process of data extraction and quality assurance is briefly described, but it may be useful to provide more details on how disagreements between reviewers were resolved and how the risk of bias assessment was conducted. This would enhance the transparency and replicability of the review.

No age inclusion criteria were set?

The authors should address the risk of bias and quality of the studies used in this systematic review.
The authors should provide the exact search strategy employed in each of the databases used. It would be advisable to include a table.

Results

Authors should check the number of articles obtained in the search in the text and in the diagram.

Authors should review Figure 1.

In the study selection section, it would be helpful to provide more information on the pre-specified criteria for exclusion, such as the definition of "populations not related" and "wrong outcome." Additionally, it would be useful to provide a justification for the decision to only include in vitro experimental studies rather than randomized controlled trials.

It would be interesting and would greatly enrich the manuscript if the authors would include in the text or in a table the vitamin D doses used and provide more specific information on the final studies of the review.

Additionally, the conclusion that vitamin D has an effect on airway remodeling is not supported by the included studies, as only two studies measured remodeling directly and the others investigated underlying mechanisms. This should be clarified in the results section.

In section 3.5, it is mentioned that the study by Plesa et al. investigated the effect of vitamin D on "human asthmatic bronchial fibroblasts", but it is not specified whether the study participants were human or whether cells from asthma patients were used. This could be clarified. It should also be referenced.

In section 3.6, TGF-β1 is mentioned as a cytokine involved in the development of "asthma and other respiratory diseases", but it is not specified what these other respiratory diseases are. It might be useful to provide a more complete list of these diseases.

Discussion

The discussion could benefit from more specific references to the studies reviewed, including authors' names and years of publication, to make it easier for readers to locate the original sources. Furthermore, while the discussion highlights the potential therapeutic role of vitamin D in asthma and other respiratory diseases, it acknowledges the complexity of the relationship between vitamin D and airway remodeling, as well as the potential inhibitory effect of TGF-β1 on vitamin D-induced airway epithelial host defense mechanisms. However, the discussion does not address the limitations or potential biases of the included studies, which could be important considerations for future research in this area. Finally, the discussion could also benefit from a more detailed analysis of the clinical implications and potential practical applications of the reviewed findings for the prevention and treatment of asthma and other respiratory diseases.

Conclusions

The authors should revise the conclusions of the study. In addition, there are some sentences that are difficult to understand and could be clearer if rewritten.

Author Response

Reviewer 2

The manuscript by Salameh et al. examines the effect of vitamin D supplementation on airway remodeling in asthma. The authors conducted a comprehensive search of electronic databases and identified experimental studies and randomized controlled trials.

The abstract could benefit from more specific information about the number of studies included in the review and the sample sizes of those studies.

Authors should rephrase the abstract. They should be clearer in the material and methods section of the abstract, indicating the databases used, study selection criteria, study population, number of articles finally included in the review.

Response 1: We thank the reviewer for pointing this out. Addendum done as suggested, kindly check lines 15-16.

It would be helpful to include any limitations or potential biases of the included studies or the review process.

Response 2: We thank the reviewer for pointing this out. Limitation added ,Kindly check line 295-299.

The conclusion could be more specific in terms of recommendations for clinical practice or future research directions.

Response 3: We thank the reviewer for the comments . We recommended in the line 249-252 & 292-294 in the discussion further studies. And as clinical implications we mentioned it in line 281-283.

Introduction.

Overall, the introduction is well written but could be improved by providing more context on the importance of airway remodelling in asthma and how it relates to the overall pathogenesis of the disease. In addition, the introduction could benefit from a clearer statement of the research question or objective of the systematic review, which could help readers better understand the purpose of the study.

Response 4: We thank the reviewer for his points. Kindly check line 59-61.

Material and methods

The process of data extraction and quality assurance is briefly described, but it may be useful to provide more details on how disagreements between reviewers were resolved and how the risk of bias assessment was conducted. This would enhance the transparency and replicability of the review.

No age inclusion criteria were set?

Response 4: We thank the reviewer for his points. The process of agreement and disagreement added .Kindly check line 85-93. Age added as well, Kindly check lines 79-80.

The authors should address the risk of bias and quality of the studies used in this systematic review.
The authors should provide the exact search strategy employed in each of the databases used. It would be advisable to include a table.

Response 5: We thank the reviewer for his points. A risk of bias assessment was carried out using cochrane risk-of-bias tool for randomized trials (RoB 2) , in accordance with the selected study criteria. The purpose of this assessment was to evaluate the quality of studies included and identify any potential risk of bias. Both authors (L.S, B.M) independently evaluated each article. Exact search strategy employed in each of the databases used added as supplementary table . Kindly check lines 85-95.

Results

Authors should check the number of articles obtained in the search in the text and in the diagram.

Authors should review Figure 1.

In the study selection section, it would be helpful to provide more information on the pre-specified criteria for exclusion, such as the definition of "populations not related" and "wrong outcome." Additionally, it would be useful to provide a justification for the decision to only include in vitro experimental studies rather than randomized controlled trials.

Response 6: Thank you for reviewer to point this out. We corrected “populations not related” and "wrong outcome." And make it more clear Kindly check lines 112-114. Regarding the justification for including in vitro studies rather than RCTs, it should be noted that all RCTs included in our systematic review did not directly measure airway remodeling. Instead, they focused on measuring the clinical manifestations of the patients. Therefore, we believe that the RCTs may not have provided an accurate representation of the real effect of vitamin D supplementation on airway remodeling.

It would be interesting and would greatly enrich the manuscript if the authors would include in the text or in a table the vitamin D doses used and provide more specific information on the final studies of the review.

Additionally, the conclusion that vitamin D has an effect on airway remodeling is not supported by the included studies, as only two studies measured remodeling directly and the others investigated underlying mechanisms. This should be clarified in the results section.

Response 7: Thank you for reviewer to point this out. vitamin D doses added in the table .

In section 3.5, it is mentioned that the study by Plesa et al. investigated the effect of vitamin D on "human asthmatic bronchial fibroblasts", but it is not specified whether the study participants were human or whether cells from asthma patients were used. This could be clarified. It should also be referenced.

Response 8: Thank you for reviewer to point this out. They are cell lines from human. The reference added.

In section 3.6, TGF-β1 is mentioned as a cytokine involved in the development of "asthma and other respiratory diseases", but it is not specified what these other respiratory diseases are. It might be useful to provide a more complete list of these diseases.

Response 9: Thank you for reviewer to point this out. The other suggestive respiratory disease added .Kindly check line 229.

Discussion

The discussion could benefit from more specific references to the studies reviewed, including authors' names and years of publication, to make it easier for readers to locate the original sources. Furthermore, while the discussion highlights the potential therapeutic role of vitamin D in asthma and other respiratory diseases, it acknowledges the complexity of the relationship between vitamin D and airway remodeling, as well as the potential inhibitory effect of TGF-β1 on vitamin D-induced airway epithelial host defense mechanisms. However, the discussion does not address the limitations or potential biases of the included studies, which could be important considerations for future research in this area. Finally, the discussion could also benefit from a more detailed analysis of the clinical implications and potential practical applications of the reviewed findings for the prevention and treatment of asthma and other respiratory diseases.

Response 10: We thank the reviewer for the comments . We recommended in the line 249-252 & 292-294 in the discussion further studies. And as clinical implications we mentioned it in line 281-283.

Conclusions

The authors should revise the conclusions of the study. In addition, there are some sentences that are difficult to understand and could be clearer if rewritten.

Response 11: We thank the reviewer for the comments . Conclusion revised.

Reviewer 3 Report

Dear Authors,

The article is very interesting. The topic of vitamin D intervention as player in asthma – associated immunomodulation is very important.  

Here are my observations.

1.    Abstract. Line 17. Please do not repeat “the studies included in this review”. I suggest before this statement to provide the number of papers or some data (core data, numbers) concerning your findings and then go to a more general statement.

2.    Abstract. With respect to vitamin D and its potential effects in asthma, there are two types of aspects: one is related to the potential correlations between vitamin D status as reflected by assessing serum 25-hydroxyvitamin D, and the other one is represented by the interventional role of offering vitamin D in order to achieve certain effects as immunomodulatory outcomes. The Abstract should clearly reflect which aspects you took into consideration.

3.    There are some typos concerning unnecessary spaces at Lines 28, 29. 75, 88, 114, 115, 139, etc.

4.    Line 88. Typo. Please use capital letter for “We attempted”..

5.    Lines 55-60. Please clearly specify the aim. The importance of such work should be discussed after Results.

6.    Methods. Please avoid “exclusion criteria included”. I suggest “Exclusion criteria were” or something similar.

7.    Line 101. Please use “out of the scope” instead of “wrong outcome”.

8.    Table 1. The reference number should be placed next to each author’s name.

9.    Table 1 . There are still some abbreviations to be explained below the table (“TSLP” + “VDR”, “IGF1”, “EGF”– please use only the abbreviation within the table and explain it below the table

10. Line 145. Please cite a paper by using only the first author’s name and “et al.” + you need to provide the reference number after “et al.” and at the end of each statement.

11. The last subsection at Discussion should be the limits/limitations of the current systematic review.

12. Conclusions. Please exclude these words which are redundant “Based on the studies mentioned, it can be concluded that”…

13. Typo. Please avoid capital letter for “vitamin” in lines 280, 283, 284, 285, etc.

Well done!

Thank you,

Dear Authors,

I introduced my editing observations above. 

Thank you

Author Response

Reviewer 3:

The article is very interesting. The topic of vitamin D intervention as player in asthma – associated immunomodulation is very important.  

Thanks for reviewer for nice comments.

Here are my observations.

  1. Line 17. Please do not repeat “the studies included in this review”. I suggest before this statement to provide the number of papers, or some data (core data, numbers) concerning your findings and then go to a more general statement.

Response 1: We thank the reviewer for pointing this out, Addendum done as suggested, kindly check lines 15-16.

  1. With respect to vitamin D and its potential effects in asthma, there are two types of aspects: one is related to the potential correlations between vitamin D status as reflected by assessing serum 25-hydroxyvitamin D, and the other one is represented by the interventional role of offering vitamin D in order to achieve certain effects as immunomodulatory outcomes. The Abstract should clearly reflect which aspects you took into consideration.

Response 2: We thank the reviewer for pointing this out. We are aiming to study the second one the interventional or supplementation role of offering vitamin D in order to prevent airway remodeling in asthmatic patients. Addendum done. Kindly check line kindly check line 14.

  1. There are some typos concerning unnecessary spaces at Lines 28, 29. 75, 88, 114, 115, 139, etc.

Response 2: We thank the reviewer for pointing this out. spaces corrected.

  1. Line 88. Typo. Please use capital letter for “We attempted”.

Response 3: We thank the reviewer for pointing this out. Typo mistake corrected.

  1. Lines 55-60. Please clearly specify the aim. The importance of such work should be discussed after Results.

Response 4: We thank the reviewer for pointing this out. Aim clarified ,Kindly check line 59-60.

  1. Please avoid “exclusion criteria included”. I suggest “Exclusion criteria were” or something similar.

Response 5: We thank the reviewer for pointing this out. Rephrasing done. Kindly check line 81.

  1. Line 101. Please use “out of the scope” instead of “wrong outcome”.

Response 6: We thank the reviewer to bring this point to notice. We corrected kindly check line 112 -114.

  1. Table 1. The reference number should be placed next to each author’s name.

Response 7: We thank the reviewer to bring this point to notice. We added the citation.

  1. Table 1 . There are still some abbreviations to be explained below the table (“TSLP” + “VDR”, “IGF1”, “EGF”– please use only the abbreviation within the table and explain it below the table

Response 8: We thank the reviewer to bring this point to notice. Abbreviation added .

  1. Line 145. Please cite a paper by using only the first author’s name and “et al.” + you need to provide the reference number after “et al.” and at the end of each statement.

Response 9: We thank the reviewer to bring this point to notice. Citation added .

  1. The last subsection at Discussion should be the limits/limitations of the current systematic review.

Response 10: We thank the reviewer to bring this point to notice. Limitation added ,Kindly check line 295-299.

  1. Please exclude these words which are redundant “Based on the studies mentioned, it can be concluded that”…

Response 11: We thank the reviewer to bring this point to notice. The word excluded.

  1. Please avoid capital letter for “vitamin” in lines 280, 283, 284, 285, etc.

Response 12: We thank the reviewer to bring this point to notice.

Round 2

Reviewer 1 Report

 .

 .

Author Response

We thank the reviewer for his/her review but we didn't find any comments for the second round, Hope our reply in the first round is satisfactory. 

Reviewer 2 Report

The authors have resolved most of the observations raised. However, some issues remain.

- The flow chart still does not match. The authors should thoroughly review the direction of arrows and numbers. The manual search results should be added to the database results, not removed. In addition, there is no concordance between the eliminated studies and the number of studies considered for the review.
- They should specify in the text the results obtained from the risk of bias of the studies. They only mention in the text that it has been performed, but they should add in the body of the text the results in a figure or table.
In addition, in the supplementary material the RoB 2 manual itself has been attached, not the result of each of the questions of the studies.

Author Response

- The flow chart still does not match. The authors should thoroughly review the direction of arrows and numbers. The manual search results should be added to the database results, not removed. In addition, there is no concordance between the eliminated studies and the number of studies considered for the review.

Response 1: We thank the reviewer for pointing this out. Apologies for the wrong direction of arrow of the search manual and the numbers search which have now been corrected. In addition, the data section in the results has been revised and corrected .

- They should specify in the text the results obtained from the risk of bias of the studies. They only mention in the text that it has been performed, but they should add in the body of the text the results in a figure or table. In addition, in the supplementary material the RoB 2 manual itself has been attached, not the result of each of the questions of the studies.

Response 1: We thank the reviewer for pointing this out. Risk of bias assessment  results were added in the manuscript as point 3.3 ( Kindly check line165-168) and the bias assessment of included studies using Revised Cochrane risk-of-bias tool for randomized trials (RoB 2) e attached as supplementary 3 figure  .
